# Investigating the Migratory Behavior of Soybean Looper, a Major Pest of Soybean, through Comparisons with the Corn Pest Fall Armyworm Using Mitochondrial Haplotypes and a Sex-Linked Marker

**DOI:** 10.3390/genes14071495

**Published:** 2023-07-22

**Authors:** Rodney N. Nagoshi, Jeffrey A. Davis, Robert L. Meagher, Fred R. Musser, Graham P. Head, Hector Portillo, Henry Teran

**Affiliations:** 1Center for Medical, Agricultural and Veterinary Entomology, United States Department of Agriculture-Agricultural Research Service, Gainesville, FL 32608, USA; rob.meagher@usda.gov; 2Department of Entomology, LSU Agricultural Center, 404 Life Science Building, Baton Rouge, LA 70803, USA; jeffdavis@agcenter.lsu.edu; 3Department of Biochemistry, Molecular Biology, Entomology, and Plant Pathology, Mississippi State University, Starkville, MS 39762, USA; fmusser@entomology.msstate.edu; 4Bayer Crop Science US, Chesterfield, MO 63017, USA; graham.head@bayer.com; 5FMC Agricultural Solutions, Stine Research Center, Newark, DE 19711, USA; hector.portillo@fmc.com; 6Corteva Agriscience™, Carr #3 Km 156.5, Salinas, PR 00751, USA; henry.teransantofimio@corteva.com

**Keywords:** soybean looper, migration, haplotypes, fall armyworm

## Abstract

The Noctuid moth soybean looper (SBL), *Chrysodeixis includens* (Walker), is an economically important pest of soybean (*Glycine max* (Linnaeus) Merrill). Because it is not known to survive freezing winters, permanent populations in the United States are believed to be limited to the southern regions of Texas and Florida, yet its geographical range of infestations annually extend to Canada. This indicates annual migrations of thousands of kilometers during the spring and summer growing season. This behavior is like that of the fall armyworm (FAW), *Spodoptera frugiperda* (J.E. Smith), also a Noctuid that is a major global pest of corn. SBL and FAW are projected to have very similar distributions of permanent populations in North America based on climate suitability modeling and the overlap in the distribution of their preferred host plants (corn and soybean). It therefore seems likely that the two species will display similar migratory behavior in the United States. This was tested by identifying genetic markers in SBL analogous to those successfully used to delineate FAW migratory pathways and comparing the distribution patterns of the markers from the two species. Contrary to expectations, the results indicate substantial differences in migratory behavior that appear to be related to differences in the timing of corn and soybean plantings. These findings underscore the importance of agricultural practices in influencing pest migration patterns, in particular the timing of host availability relative to mean seasonal air transport patterns.

## 1. Introduction

Corn (*Zea mays* Linnaeus) and soybean (*Glycine max* (L.) Merr.) represented the two highest acreage field crops in the United States in 2022, with both estimated at nearly 90 million acres according to the National Agricultural Statistics Service [1]. They largely overlap in geographical distribution and are frequently rotated at 2–4-year intervals in the same fields to preserve soil quality and to increase yields [2]. While herbivorous insect population behaviors are influenced by many factors [3], the availability of plant hosts is likely to be among the most significant in defining the timing, direction, and magnitude of their local dispersion and longer migratory movements. Therefore, the similarity in the geographical distribution of corn and soybean in the U.S. suggests that pests with similar physiological characteristics targeting these crops will display comparable seasonal migration behavior.

Two Noctuid moth species, the soybean looper (SBL), *Chrysodeixis includens* (Walker), and fall armyworm (FAW), *Spodoptera frugiperda* (J.E. Smith), are important pests of soybean and corn, respectively. They are sufficiently similar in physiology to give nearly identical climate suitability maps for permanent populations by CLIMEX modeling (compare [4] and [5]). These projections indicate that in the United States, permanent SBL and FAW populations are primarily limited to southern Florida and southern Texas, where mild winter conditions allow for continuous breeding [6]. Both species are reported to have a broad host range with substantial overlap in their secondary hosts, including the capacity to develop on the primary host of the other [4,7]. Their geographical range is extensive and overlapping, with infestations extending from the overwintering sites in FL and TX to southern Canada.

These observations indicate that SBL and FAW annually undergo a northward migration in the United States that extends over several thousand kilometers, covers the same geography, occurs during the same U.S. spring–summer growing period, and is therefore subject to the same set of seasonal wind vectors. Given these similarities, it seems likely that the migratory pathways and behaviors of SBL will resemble those of FAW.

The migratory behavior of North American FAW populations has been extensively studied. Large populations emerge from southern Florida and Texas in the early spring and move northward in a stepwise pattern over 2–3 generations that corresponds to the northward progression of corn planting facilitated by strong northerly seasonal winds [8]. The large size of the FAW migrating population is suggested by several observations that include the radar imaging of high-density moth flights and the extrapolation of FAW density from bat stomach contents that had been feeding at high altitudes [9,10,11,12]. Modeling verified by genetic studies demonstrated that the Texas and Florida wintering populations consistently follow distinct migratory pathways, with Texas the source of infestations in the central U.S. in states west of the Appalachian Mountain range while Florida migrants are largely limited to the Atlantic coastal states as far north as Maryland and New Jersey [13,14,15]. The mixing of the two wintering populations is largely restricted to the southeastern and northeastern regions of the U.S., with the Appalachian Mountains potentially acting as a physical barrier that impedes more extensive interactions. As a result, the Texas and Florida wintering populations have limited interactions, the degree of which depends on the magnitude of return migrations from the sites of mixing. This has yet to be quantified and could vary substantially from year to year depending on wind patterns and agricultural practices.

In contrast, remarkably little is known about SBL migration in North America, especially given the economic importance of this pest. Research has been limited to field surveys and trapping studies that found SBL continuously in Texas and Florida but not in more northern states during the winter months [16]. The magnitude, timing, and the specific directions of the migrations from the overwintering locations remained largely uncharacterized [17]. This lack of migratory information is particularly consequential because SBL exhibits resistance to many broad spectrum insecticides that is often variable depending on time and location [18,19,20]. A better description of SBL migration patterns is needed to identify the source populations of these resistance traits and predict their distribution during the annual North American migration.

The objective of this study was to compare the described the migratory behavior of SBL using the genetic strategy that successfully delineated the FAW migratory pathways. This methodology was designed to accommodate the specific challenges presented by migratory moths, most notably the extensive opportunities for the mixing of populations and consequent cross-hybridization [21]. As a result, the geographical isolation of migratory populations as identified by population-specific genetic markers will likely be limited in scope and variable over time. These challenges were overcome for FAW by identifying genetic differences between populations that are resilient to moderate levels of gene flow. Specifically, both Texas and Florida FAW shared the same set of mitochondrial haplotypes but differed in their relative proportions, with the differences consistently observed over multiple years and collection sites [22]. Apparently, there is sufficient gene flow to homogenize the FL and TX populations with respect to the presence of the more common mitochondrial haplotypes but not enough to eliminate differences in their proportions.

The methodology used to study FAW migration was applied to a set of SBL collections from multiple locations in North America and Puerto Rico. The SBL specimens were characterized for two segments of the mitochondrial *Cytochrome oxidase* subunit I (*COI*) gene, one commonly used for DNA barcoding [23], and the other that identified haplotypes used to differentiate geographical populations of FAW [24]. One exon and one intron segment from the Z-chromosome-linked *Triosephosphate isomerase* (*Tpi*) gene were similarly analyzed. Polymorphisms in the *Tpi* gene have been found that can distinguish between subpopulations of Noctuid moth species, including the host strains of *S. frugiperda* [25]. We report evidence of differences in migratory behavior between FAW and SBL that could be related to differences in the availability of their primary host plants.

## 2. Materials and Methods

### 2.1. Specimen Collections

Specimens were collected from soybean in field plots either as larvae or by pheromone trapping (Table 1). The larvae were reared to produce laboratory colonies, with specimens obtained after the first (F1) or second (F2) generation for molecular analysis. The collections from pheromone traps were made up of adult males that were stored in ethanol and initially identified as soybean looper by morphological criteria before molecular analysis.

### 2.2. DNA Preparation

A preparation containing both genomic and mitochondrial DNAs was purified from individuals as previously described [26] and stored in TE (10 mM Tris [pH 8], 1 mM EDTA) at −20 °C. Additional preparations were performed using a simpler variation of the described method. Specimens were homogenized in 0.8 mL of phosphate-buffered saline (PBS, 20 mM sodium phosphate, 150 mM NaCl, pH 8.0) in a 7 mL Dounce homogenizer. The preparation was transferred to 2.0 mL microcentrifuge tubes and the cells/tissue were pelleted by centrifugation at 12,000× *g* for 3 min at room temperature. The pellet was resuspended in 400 µL of Genomic Lysis Buffer (Zymo Research, Irvine, CA, USA), transferred to a 1.5 or 2.0 mL microcentrifuge tube and incubated at 55 °C for 10 min. The preparation was pelleted using centrifugation at 12,000× *g* for 10 min at room temperature. The supernatant was transferred to a Zymo-Spin III column (Zymo Research, Orange, CA, USA) and processed according to manufacturer’s instructions. The DNA preparation was increased to a final volume of 100 µL with distilled water. Genomic DNA preparations of soybean looper samples were stored at −20 °C and analyzed as needed.

### 2.3. Isolation of the DNA Segments by Polymerase Chain Reaction (PCR)

The relevant segments of the SBL *COI* gene sequence were identified aligning the well-characterized FAW *COI* sequences with the SBL mitochondria genome sequence (GenBank LR797832). Four primers were generated from the identified SBL *COI* locus, sC101f (5′-TCGAGCAGAATTAGGTACCCC-3′), sC678r (5′-ATAGGATCTCCTCCTCCTGCT-3′), sC891f (5′-TACACGAGCTTATTTTACTTC-3′), and sC1457r (5′-ATATCATTCAATAGAAGAGG-3′).

The SBL COIA segment (sCOIA) and the downstream sCOIB segments were amplified via polymerase chain reaction (PCR) amplification using the primer pairs sC101f/sC678r and sC891f/sC1457r, respectively (Figure 1A). The PCR amplification was performed in a 30 µL reaction mix containing 3 µL 10X manufacturer’s reaction buffer, 1 µL 10 mM dNTP, 0.5 µL 20 µM primer mix, 1 µL DNA template (between 0.05–0.5 µg), and 0.5 unit Taq DNA polymerase (New England Biolabs, Beverly, MA, USA). The thermocycling program was 94 °C (1 min), followed by 33 cycles of 92 °C (30 s), 56 °C (30 s), 72 °C (30 s), and a final segment of 72 °C for 3 min. Typically, 96 PCR amplifications were performed at the same time using either 0.2 mL tube strips or 96-well microtiter plates.

The SBL *Tpi* gene segment was isolated by a partially nested PCR amplification procedure. The first reaction used FAW primers fTpi208f (5′-TACAAAGCATTGTACCACCCTC-3′) and fTpi1195r (5′-AGTCACTGACCCACCATACTG-3′) using the same protocol as described for the *COI* gene on SBL genomic DNA, but with an annealing temperature of 52 °C. The second PCR was performed with FAW primers fTpi242f (5′-CGCACAAAACTGCTGGAAG-3′) and fTpi1195r on 1 µL of the first reaction. fTpi242f lies within the fragment amplified by sTpi86f/sTpi1195r (Figure 1B). DNA sequence information from the final amplification product was used to generate the SBL-specific primers sTpi208f (5′-GGTGAAATCTCTCCAGCCATG-3′), sTpi412f (5′-ATGGCCTGAAAGTCATTGCCTG-3′), and sTpi1140r (5′-GCAGACACATTCTTAGCCAGCC-3′, Figure 1). Subsequent PCR reactions using SBL-specific primers were performed at the annealing temperature of 56 °C. All primers used for PCR and DNA sequencing were synthesized by Integrated DNA Technologies (Coralville, IA, USA).

For fragment isolations, 6 µL of 6× gel loading buffer was added to the amplification reaction and the entire sample run on a 1.8% agarose horizontal gel containing GelGreen (Biotium, Hayward, CA, USA) in 0.5× Tris-borate buffer (TBE, 45 mM Tris base, 45 mM boric acid, 1 mM EDTA pH 8.0). Fragments were visualized on a Blue LED transilluminator (Thermo Fisher Scientific, Waltham, MA, USA) and cut out from the gel. Fragment isolation was performed using Zymo-Spin I columns (Zymo Research, Orange, CA, USA) according to the manufacturer’s instructions.

### 2.4. Characterization of Haplotypes

Each specimen was anticipated to have a single *COI* haplotype that is maternally inherited. To simplify the analysis of sCOIB, haplotypes were defined by a subset of SNPs chosen based on a polymorphism frequency greater than 10% across all collections. This criterion was satisfied by sites sc1035 and sc1272 that were each polymorphic for C/T and produced four observed haplotypes (C_1035_C_1272_, C_1035_T_1272_, T_1035_C_1272_, and T_1035_T_1272_). The FAW COIB haplotypes were similarly defined, using two SNPs that are also C/T polymorphic, producing the FAW COI-CSh haplotypes of C_1164_C_1287_ (h1), C_1164_T_1287_ (h2), T_1164_C_1287_ (h3), and T_1164_T_1287_ (h4). The relative proportions of the two most common fCOIB haplotypes are described by the simple metric (M) of M_FAW_ = (h4 − h2)/(h2 + h4) (refs). The sCOIB haplotypes are similarly described, M_SBL_ = (C_1035_T_1272_ − T_1035_C_1272_)/(C_1035_T_1272_ + T_1035_C_1272_).

The *Tpi* hapotypes are more complicated as the location of the *Tpi* gene on the Z-chromosome means that two gene copies are present in male (Z/Z) and so heterozygous allele combinations are possible, while females (Z/W) carry only a single copy. Because DNA sequencing is performed on PCR fragments directly produced from the specimen, heterozygosity for the Tpi segment will result in the simultaneous sequencing of two different alleles. This will produce chromatographs with overlapping curves at the sequence mismatch. Particularly disruptive are heterozygous indels (insertions and deletions) because these can shift the DNA sequencing frame of the two DNA strands, resulting in the continued mismatch downstream of the indel. All the indels in this study were limited to the Tpi intron sequences and so did not impact data from the Tpi exon. *Tpi* allele frequencies were calculated as per specimen (rather than per chromosome).

### 2.5. DNA Sequence Analysis, Statistics, and Data Availability

The isolated fragments were analyzed by DNA sequencing using the appropriate primers (Azenta, South Plainfield, NJ, USA). DNA comparisons and alignments were performed using the Geneious Pro 10.1.2 program [27]. All haplotypes obtained in this study have been deposited in GenBank. These include the sCOIA (OR129847-OR129855), sCOIB (OR128567-OR128591), and sTpiE (OR124064-OR124071) variants. Representative sequences of i65del_Tpi_ (OR124276) and i65+_Tpi_ (OR124332) were also deposited. Statistical analyses were conducted using GraphPad Prism version 7.00 for Mac (GraphPad Software, La Jolla, CA, USA). The generation of graphs was performed using Excel and PowerPoint (Microsoft, Redmond, WA, USA). The sCOIB phylogenetic tree was produced using PhyML 3.3.20180214 [28,29]. The analysis underwent bootstrap testing (100 replicates) with the optimal tree shown and drawn to scale. The evolutionary distances were computed using the maximum composite likelihood method [30] and diagrammed based on midpoint rooting with the evolutionary analysis conducted using Geneious Pro 10.1.2.

## 3. Results

### 3.1. Characterization of SBL Genetic Markers

The first objective was to find highly polymorphic genetic markers in SBL for use in detecting genetic differentiation among populations. The markers should be able to produce multiple haplotypes with high enough frequencies to make possible the differentiation of populations without requiring a prohibitively large sample size. In an initial survey, SBL specimens from six southeastern states in 2016 were analyzed for three DNA segments, two from the mitochondrial *COI* gene (sCOIA, sCOIB), and one from an exon (sTpiE) of the Z-linked *Tpi* gene (Figure 1A,B). All three segments were from the presumptive coding region of either the predicted *COI* or *Tpi* products, and all detected polymorphisms that were single base substitutions did not alter the amino acid sequence. The overall results were consistent with findings from Brazilian SBL populations, indicating low genetic variability for the species [31,32].

Eight SNPs were found in the SBL COIA (sCOIA) segment from 332 samples analyzed, with six found only once (singletons). The remaining two SNPs (sCOIA_474_ and sCOIA_528_) showed weak polymorphism with at least 98% of specimens expressing one type (Table 2). Maximum likelihood phylogenetic analysis confirmed the SBL identity of the collections, with the nine observed haplotypes clustering with GenBank sequences identified as SBL (Figure 2A). Low variability was also observed with the SBL *Tpi* exon segment, sTpiE. The four most variable SNPs had polymorphism frequencies of no more than 10% (Table 2). These produced eight haplotypes. Sequences for the SBL *Tpi* gene were not available in GenBank, but the observed sTpiE haplotypes clustered together and were separated from FAW and other selected species via maximum likelihood phylogenetic comparisons (Figure 2B).

Greater variability was observed with the sCOIB segment that lies downstream of sCOIA. Nine polymorphic sites were identified, with seven found in no more than two specimens. Two sites, sCOIB_1035_ and sCOIB_1272_, had polymorphism frequencies of about 30% (Table 2). The analysis of these SNPs was expanded to include SBL collections from 2017 to 2022, with the observed sCOIB_1035_ and sCOIB_1272_ combinations color-coded in Figure 3A. In the expanded sample set, 71% were of the C_1035_T_1272_ haplotype category and 26% were T_1035_C_1272_ (Figure 3B). The distribution of the four haplotypes was significantly nonrandom (*p* < 0.0001, *Chi*-square = 921.2, *df* = 3). Phylogenetic analysis using maximum likelihood demonstrates that the four COIB_SBL_ combinations cluster separately, with the C_1035_C_1272_ specimens appearing to be more closely related to the majority C_1035_T_1272_ group, while T_1035_C_1272_ and T_1035_T_1272_ group separately (Figure 3C).

### 3.2. Comparisons between the SBL and FAW COIB Haplotype Profiles

The sCOIB frequency data were organized by state to provide an estimate of the geographical distribution of the four haplotype classes (Figure 4A). All collections were dominated by the C_1035_T_1272_ and T_1035_C_1272_ populations, which together made up at least 90% samples from each state. The rare C_1035_C_1272_ haplotype was only found in FL and TX. The T_1035_T_1272_ haplotype was more broadly detected and was about three-fold more frequent than C_1035_C_1272_.

The COIB_SBL_ data were compared to the distribution of FAW COIB haplotypes (fCOIB) summarized from previous studies (Figure 4B) [14,22]. The fCOIB haplotypes also fell into four categories (CSh1–4) with two, CSh2 and CSh4, predominant in all collections but varying in their relative proportions [22]. The proportions of CSh2 and CSh4 were described by the metric, (CSh4 − CSh2)/(CSh2 + CSh4) = M_FAW_, where the values are in frequencies. An analogous metric, M_SBL_ = (C_1035_T_1272_ − T_1035_C_1272_)/(C_1035_T_1272_ + T_1035_C_1272_), was used to quantify the sCOIB haplotype differences.

These M_FAW_ and M_SBL_ values were analyzed in two ways based on geography. The first separately compares the two species in states along the eastern coast versus those that lie west of a line approximated by the Appalachian Mountain range. This assesses variations that result from the two migratory sources of FL and TX based on migration pathways identified for FAW [14,15]. The second compares states that border the Gulf of Mexico, which encompasses the region capable of supporting overwintering populations, with the more northern collections that are dependent on long-distance migration. This examines how well the migratory populations retain the genetic composition of the source populations.

As expected from previous studies, the mean M_FAW_ of the collections from the eastern coast was significantly different from that of the more western states, reflecting the different M_FAW_ means of the FAW that overwinter in FL and TX (FAW East vs. West, Table 3). In comparison, the analogous M_SBL_ metric for SBL was not significantly different for comparisons of the equivalent groups of collections (SBL East vs. West, Table 3). This finding is consistent with the absence of clear differences between the SBL populations that overwinter in FL and TX.

The same dataset was regrouped to test the persistence of the TX and FL haplotype proportions during migration. With FAW, no significant difference was found in the mean M_FAW_ of the states that border the Gulf of Mexico compared to the more northern states (FAW North vs. South, Table 3), which is consistent with the supposition that FAW migration involves the movement of large enough numbers to maintain the genetic constitution of the source populations. In contrast, a significant difference was found between the mean M_SBL_ for the equivalent north and south SBL groupings (SBL North vs. South, Table 3).

A potentially complicating factor in the SBL analysis is that specimen collections were a mix of field collections and the first or second generation of progeny from colonies derived from larval field collections (Table 1). To test whether the limited artificial rearing significantly altered the haplotype profiles, the M_SBL_ of the field and colony collections were calculated and compared by two-tailed *t*-test analysis. The differences among the M_SBL_ means of the field (0.47 ± 0.32) and colony (0.38 ± 0.50) collection strategies were not statistically significant (*p* = 0.6609; *t* = 0.4453; *df* = 20).

### 3.3. Using an Intron Segment from Tpi as a Genetic Marker

To confirm the sCOIB results, we developed a nuclear genetic marker derived from an intron segment located immediately adjacent to the previously analyzed *Tpi* exon segment. In FAW, this intron exhibits a high frequency of indels (insertions and deletions) and SNPs [33,34], which was also observed in the corresponding SBL *Tpi* intron segment (designated sTpi140). This fragment extends from the 5’ splice site to approximately 140-bp into the intron and includes a 7-bp deletion, i65del_Tpi_ (Figure 1B), which is readily distinguishable by DNA sequencing from the non-deleted alternatives, i65(+)_Tpi_. The i65del_Tpi_ allele showed substantial variation among collections with a range of 0% to 47% and a mean of 16 ± 14% (Figure 4C). All collections were similar in having a majority i65(+)_Tpi_ composition; however, like T1035_COIB_, the mean frequency of i65del_Tpi_ in the more northern states of SC, NC, VA, TN, KS (32 ± 14%) was significantly higher (two-tailed *t*-test: *p* = 0.0006, *t* = 4.314, *df* = 15) than the 10 ± 7% mean found for the states bordering FL, TX, or the Gulf of Mexico.

## 4. Discussion

In this study, the *COI* haplotype distribution in North America was compared between FAW and SBL to investigate the migratory behavior of the latter species. With FAW, regional differences were consistently observed that corresponded to distinct migratory pathways, one originating in FL and the other from TX. The *COI* haplotype profiles at the migratory destinations were indistinguishable from the source populations, consistent with the northward FAW migration involving populations large enough to generally maintain the relative haplotype proportions. Modeling studies project that this behavior is dependent on the coincidence of seasonal wind vectors favorable to northward air transport with large acreages of corn that shifts northward as the growing season progresses [8]. This pattern of corn agriculture provides a surplus of plant hosts to support and amplify high density migratory populations.

The SBL COIB haplotype distribution pattern differed from FAW in two respects. First, there was no clear distinction between the TX and FL populations, which prevents the use of these markers to study differential migration from these presumptive overwintering locations. This is not surprising because such allele differences (assuming neutral selection) occur by chance and can only be maintained if gene flow is sufficiently restricted. Second, evidence of significant haplotype differences was observed between the southern and northern collections, a finding supported by the data from the *Z*-chromosome i65del_Tpi_ allele. This tendency for the samples from the more northern migratory destinations to genetically differ from the presumed migratory sources could be explained if SBL migrations involved smaller populations that were more frequently subject to bottleneck and founder effects. One plausible explanation for why this might occur without having to invoke innate differences in migratory behavior with FAW has to do with differences in the agriculture of soybean, the primary host for SBL.

Although corn and soybean plantings geographically overlap, they differ temporally in availability. Soybean is typically planted after corn for economic reasons, with an average difference of about 40 days for earliest planting (Figure 5A). A more detailed comparison is provided by National Agriculture Statistical Service data for GA that should approximate crop phenology for the southeastern region. It shows that most corn had emerged by the end of April (Figure 5B). If FAW laid eggs at this time, the adults would be ready to migrate in May and June when northerly wind vectors are strong for most of the region (Figure 5C). The combination of high-density populations and strong winds result in the long-distance migration of groups large enough to consistently reflect the genetic composition of the source populations. Consequently, the haplotype proportions at the migratory destinations will be similar to, and thereby identify, the originating source (Figure 6).

In contrast, most GA soybean was planted in May and emerged in June. This means that the migratory SBL in these early growing season months will have developed on secondary hosts. These non-optimal conditions will likely produce relatively small populations that may only be inconsistently representative of the source populations (Figure 6). In addition, smaller migratory groups are more likely to undergo bottleneck events that can further alter their haplotype composition. By the time soybean acreage increases to levels that can support larger and more representative SBL populations, the substantially weaker seasonal wind vectors over most of the southeast will limit their northward expansion (July and August, Figure 5C). In summary, we conclude that early season SBL migration will involve smaller groups than FAW, while the later emerging large SBL populations in the overwintering and adjacent southeastern states will remain largely localized due to weaker and less organized wind vectors. As a result, SBL is expected to show more variability in haplotype composition and proportions with increasing migratory distance.

While we believe the above represents the simplest explanation for the observed genetic structure, we certainly cannot preclude other possible influences. The most likely alternative in our opinion is the existence of multiple SBL populations that differ in migration behavior or fitness, with the latter assuming different selection pressures in the more northern states from those nearer the overwintering locations. Genetic evidence for two SBL populations has been reported from Brazil. One study using inter-simple sequence repeat (ISSR) markers found low but significant genetic differentiation that was attributed to a possible recolonization of Brazil [31]. The two groups exhibited substantial gene flow and no genetic structure, with the study based on a single year that precluded replication of the observation. In the second report, mitochondrial haplotypes and nuclear markers detected genetic differentiation between SBL collected from soybean and cotton, suggesting a possible host-based divergence. However, this differentiation was not observed in similar surveys at another location [32]. These suggestions of two SBL populations are at best preliminary and even if correct, there is no indication that they differ in characteristics that would explain the observed population structure in this study. In this regard, we do have evidence from our genetic data for two SBL populations in the United States that will be presented in a companion paper. Whether these differ in ways relevant to migration remains unclear.

The degree to which SBL populations in Puerto Rico contribute to infestations in the continental U.S. has long been speculated, particularly with the discovery of diamide pesticide resistance in Puerto Rico and periodic reports of diamide failures in the U.S. [35]. In this study, the Puerto Rico haplotype proportions are generally like those seen in the continental U.S. except for the sCOIB haplotypes in 2019 and 2020. The T_1035_C_1273_ sCOIB haplotype was found in all the US collections during all time periods but was not detected in Puerto Rico during those two years. This could indicate that U.S. populations are not consistently entering Puerto Rico in large numbers, but more extensive surveys are needed. The level of northward migration from Puerto Rico remains unclear.

In conclusion, the haplotype proportion method was able to detect population structure for SBL in the U.S., with the southern populations differing in the ratios of both *COI* and *Tpi* marker frequencies compared to more northern collections. This can be explained by the timing of soybean planting, which leads to high-density SBL populations emerging when migratory conditions in the form of wind vectors are not favorable for much of the U.S. The hypothesis is that this difference in planting times between corn and soybean leads to different migratory behaviors for FAW and SBL, despite their physiological similarities.

## Figures and Tables

**Figure 1 genes-14-01495-f001:**
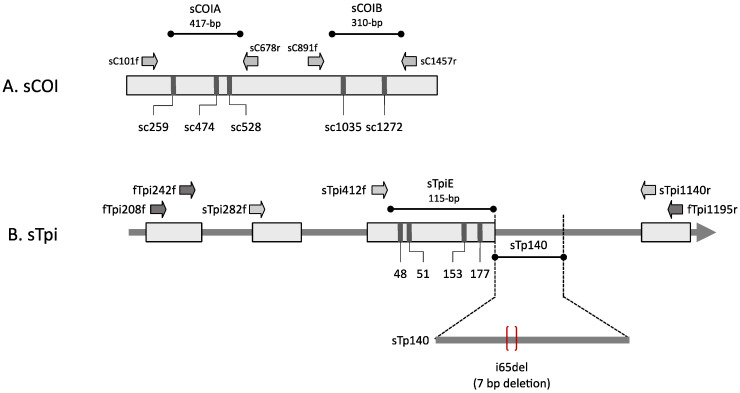
Diagrams of the SBL COI (sCOI) and SBL Tpi (sTpi) loci identifying the locations of SBL SNPs and the i65del_Tpi_ marker. Block arrows identify primers used for PCR amplification and DNA sequencing, with dark-filled arrows indicating FAW primers and light-filled arrows indicating SBL primers. (**A**) The sCOI locus with the fragments sCOIA and sCOIB are indicated. (**B**) The sTpi locus shows the relative locations of sTpiE, sTpi140, i65del_Tpi_. Image is not to scale.

**Figure 2 genes-14-01495-f002:**
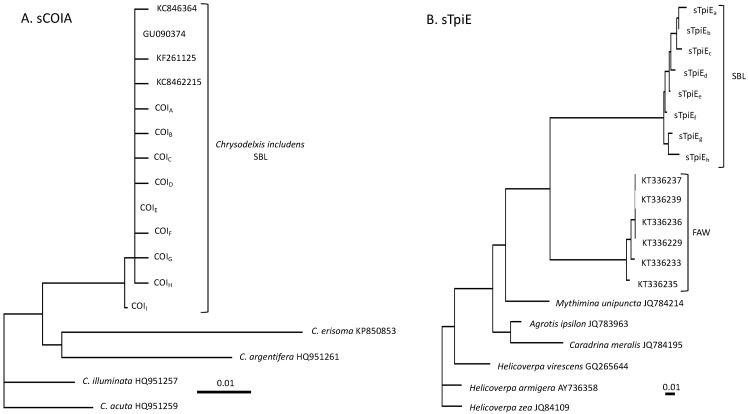
Comparisons of observed sCOIA and sTpiE sequences with selected GenBank sequences via phylogenetic analysis. Trees were created via maximum likelihood with 100 repetition bootstrapping. (**A**) Phylogenetic tree with the nine sCOIA haplotypes (COI_A–I_) found in this study. GenBank sequences KC846364, GU090374, KF261125, and KC8462215 are identified as *Chrysodelxis includens*. (**B**) Phylogenetic tree of the sTpiE haplotypes from this study (sTpi_a–h_).

**Figure 3 genes-14-01495-f003:**
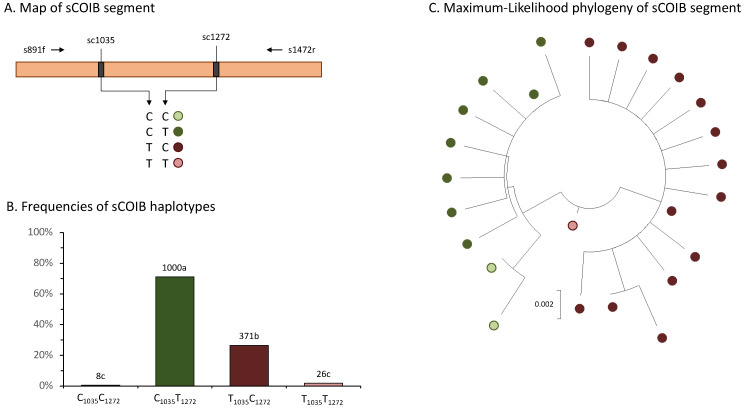
Information about the sCOIB SNPs. (**A**) Diagram of the sCOIB fragment with horizontal arrows indicating primers used for PCR amplification and DNA sequencing. The locations of the two major sCOIB SNPs are indicated, with the observed polymorphisms listed below. The pairings represent haplotypes and are color-coded for the rest of the figure. (**B**) Bar-graph shows haplotype frequencies. The numbers above the columns indicate the number of specimens expressing that haplotype. Different lower case letters indicate statistically significant difference in the means by ANOVA analysis and Tukey’s multiple comparisons test (*p* < 0.0001, *F* = 103.5, *r*^2^ = 0.7871). (**C**) The maximum likelihood phylogeny of the sCOIB variants observed.

**Figure 4 genes-14-01495-f004:**
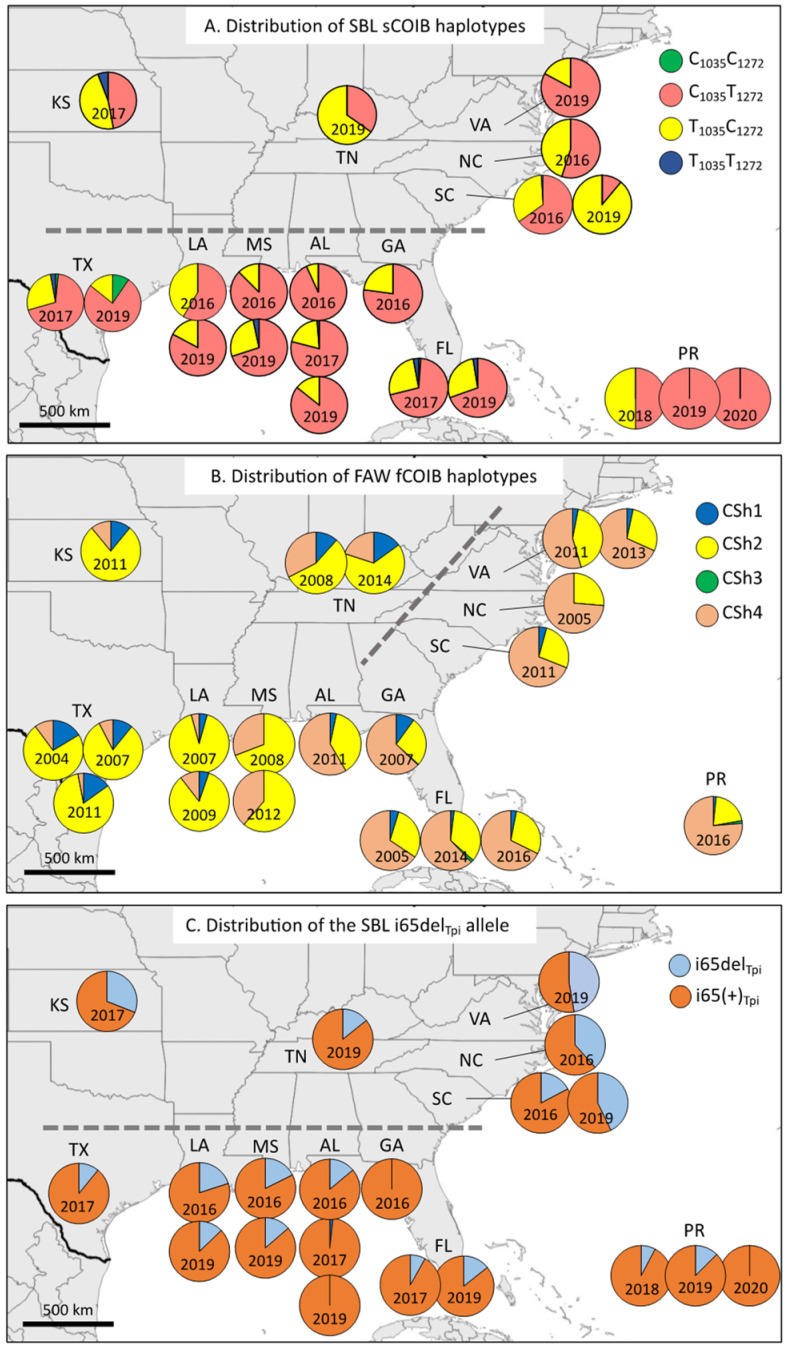
Geographical map of the distribution of haplotype proportions by state. Each pie chart represents pooled collections from the state for a given year. A horizontal, dashed grey line separates states bordering the Gulf Coast and FL from the more northern migratory destinations. The diagonal grey line approximates the location of the Appalachian Mountain range and the approximate border separating FAW migrating from TX from those originating from FL. (**A**) The distribution of the SBL sCOIB haplotypes. (**B**) The distribution of the FAW fCOIB haplotypes. (**C**) The distribution of the SBL i64del_Tpi_ marker.

**Figure 5 genes-14-01495-f005:**
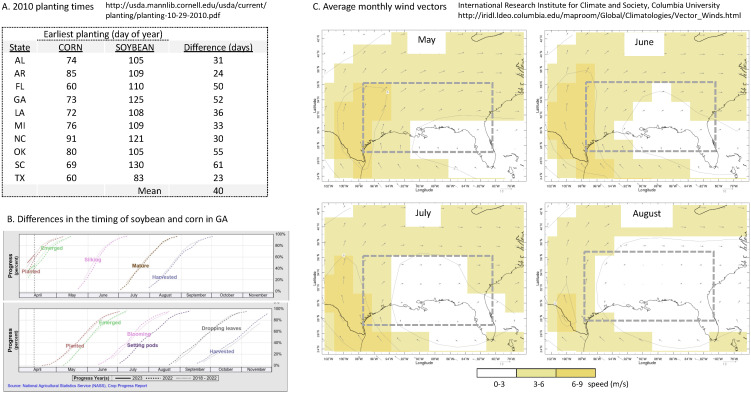
Corn and soybean phenology data and mean seasonal wind vectors for selected portion of the U.S. (**A**) earliest plant dates for 2010 from the USDA ESMIS system at Cornell University. (**B**) crop phenology dates for GA from the National Agriculture Statistics Service. (**C**) Mean monthly wind vectors for May–August for the southeastern U.S. from the International Research institute for Climate and Society at Columbia University. Dashed box outlines region where wind vectors change substantially during the spring-summer growing season.

**Figure 6 genes-14-01495-f006:**
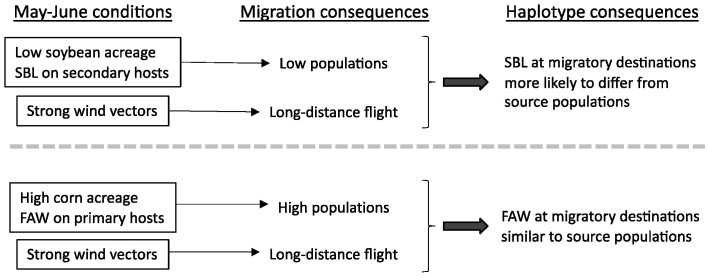
Summary of relevant conditions and predicted consequences for the migration of FAW and SBL during the early U.S. growing season.

**Table 1 genes-14-01495-t001:** Source information for SBL collections. All collections from soybean.

Collection	State	Year	Type	Collector
AL2016	Alabama	2016	F1 larva	J. Davis
AL2017	Alabama	2017	Pheromone trap	R. Meagher
AL2019	Alabama	2019	F1–2 larva	F. Musser, B. Catchot
FL2017	Florida	2017	Pheromone trap	R. Meagher
FL2019	Florida	2019	Pheromone trap	R. Meagher
GA2016	Georgia	2016	F1 larva	J. Davis
KS2017	Kansas	2017	Pheromone trap	B. McCornack
LA2016	Louisiana	2016	F1 larva	J. Davis
LA2019	Louisiana	2019	F1–2 larva	F. Musser, B. Catchot
MS2016	Mississippi	2016	F1 larva	J. Davis
MS2019a	Mississippi	2019	Pheromone trap	R. Meagher
MS2019b	Mississippi	2019	F1–2 larva	F. Musser, B. Catchot
NC2016	N. Carolina	2016	F1 larva	J. Davis
PR2018	Puerto Rico	2018	larva from field	H. Portillo
PR2019	Puerto Rico	2019	larva from field	H. Teran
PR2020	Puerto Rico	2020	F1–2 larva	F. Musser, B. Catchot
SC2016	S. Carolina	2016	F1 larva	J. Davis
SC2019	S. Carolina	2019	F1–2 larva	F. Musser, B. Catchot
TN2019	Tennessee	2019	F1–2 larva	F. Musser, B. Catchot
TX2017	Texas	2017	Pheromone trap	R. Parker
VA2019	Virginia	2019	F1–2 larva	F. Musser, B. Catchot

**Table 2 genes-14-01495-t002:** Selected SNPs in the SBL *COI* (sCOIA, sCOIB) and SBL *Tpi* exon (sTpiE) segments. Frequencies are for the combined 2016 collections from AL, GA, MS, LA, and SC.

SNP	*n*	Polymorphism	Frequency
sCOIA_474_	332	C/T	99% C
sCOIA_528_	332	C/T	98% T
sCOIB_1035_	316	C/T	67% C
sCOIB_1272_	316	C/T	68% T
sTpiE_48_	249	A/G	97% A
sTpiE_51_	242	A/G	90% G
sTpiE_153_	259	C/T	97% C
sTpiE_177_	264	C/T	97% C

**Table 3 genes-14-01495-t003:** Statistical comparisons of the regional distribution of COIB haplotypes in FAW and SBL collections by unpaired two-tailed *t*-test analysis.

	Fall Armyworm (FAW)				Soybean Looper (SBL)			
Region	East ^1^	West ^2^	North ^3^	South ^4^	East ^1^	West ^2^	North ^3^	South ^4^
Mean M ^5^	0.37	−0.53	−0.09	−0.18	0.29	0.46	0.00	0.56
sd	0.11	0.34	0.50	0.55	0.47	0.36	0.50	0.18
	East vs. West		North vs. South		East vs. West		North vs. South	
*p*	<0.0001		0.7306		0.3871		0.0017	
*t*	7.180		0.3500		0.8876		3.715	
*df*	17		16		17		17	

^1^ FL, GA, SC, NC, VA. ^2^ TX, LA, MS, KS, TN, AL. ^3^ KS, TN, VA, NC, SC. ^4^ TX, LA, MS, GA, FL, AL. ^5^ M_FAW_ = (CSh4 − CSh2)/(CSh2 + CSh4), M_SBL_ = (C_1035_T_1272_ − T_1035_C_1272_)/(C_1035_T_1272_ + T_1035_C_1272_).

## Data Availability

Relevant DNA sequence data are openly available in GenBank. All other data are present in this study.

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
