# Peer review of "Investigating the Migratory Behavior of Soybean Looper, a Major Pest of Soybean, through Comparisons with the Corn Pest Fall Armyworm Using Mitochondrial Haplotypes and a Sex-Linked Marker"

_genes, 2023, doi:10.3390/genes14071495_

Round 1

Reviewer 1 Report

This is my first review of the manuscript " genes-2476990- Investigating the migratory behavior of soybean looper, a major pest of soybean, through comparisons with the corn pest fall armyworm using mitochondrial haplotypes and a sex- linked marker " submitted to genes. The soybean looper (SBL), Chrysodeixis includens (Walker), and fall armyworm (FAW), Spodoptera frugiperda (J.E. Smith) display comparable seasonal migration behavior in the United States. However, the results indicate substantial differences in migratory behavior that appear to be related to differences in the timing of corn and soybean plantings. This is an important and meaningful research.

Overall, the text is well written. However, I'm a little confused about how to use genetic marker from SBL to extend to FAW. There is insufficient evidence of to establish a link between these two pests. In my opinion, the methods used to link the migration between SBL and FAW seem farfetched. Please explain reasonably before the manuscript can be considered further.

Author Response

<<<However, I'm a little confused about how to use genetic marker from SBL to extend to FAW. There is insufficient evidence of to establish a link between these two pests. In my opinion, the methods used to link the migration between SBL and FAW seem farfetched. Please explain reasonably before the manuscript can be considered further.>>>

We are not trying to establish a link between SBL and FAW. We are using a strategy successfully applied to FAW to now study SBL. Specifically, genetic markers based on polymorphisms in the COI and Tpi genes were previously used with great success to describe FAW migration patterns and population distributions in North America as described in lines 71-87 and 124-125. The advantages of the FAW methodology to study populations with significant gene flow are described in lines 105-117. In this paper we used the same strategy to study SBL populations in North America, as stated in lines 118-127.

So just to emphasize, we are not linking the migration of SBL and FAW. We are simply comparing the migration patterns of these two species. This is stated in lines 396-407.

Reviewer 2 Report

I found some minor mistakes in this manuscript. Please see below and carefully revise your manuscript. A revision is needed.

Corn (Zea mays L.) and soybean (Glycine max (L.) Merr.) represented the two highest acreage crops in the United States in 2022 ? Among field crops or among all agricultural crops? Please clarify it.

They are sufficiently similar in such physiological metrics as temperature tolerance, dry/wet sensitivity, and degree-day developmental profiles to give nearly identical climate suitability maps by CLIMEX modeling [4]: only this ref. 4, is not enough to declare above sentence. Please use appropriate reference.

g [4]. These projections indicate that in the United States permanent SBL and FAW populations: check the space

Is fall armyworm can overwinter in the TX? Which stage of FAW overwinter in the Tx and Canada?

FAW density from bat stomach contents that had been feeding at high altitudes [7-10]. Modeling verified by genetic studies demonstrated that the Texas: check the space

Mountains potentially acting as a physical barrier that impedes more extensive interactions. As a result, the Texas and Florida wintering populations have limited interactions,: check the space

I found many writing errors in the manuscript, particularly with the space. Please check carefully and correct accordingly.

Cytochrome oxidase subunit I, Triosephosphate isomerase: gene name should be italicized

Why author used old samples? If possible, please compare with latest samples

Please compare the mean differences for Figure 3B.

Author Response

Corn (Zea mays L.) and soybean (Glycine max (L.) Merr.) represented the two highest acreage crops in the United States in 2022 ? Among field crops or among all agricultural crops? Please clarify it.

 Done (line 47)

They are sufficiently similar in such physiological metrics as temperature tolerance, dry/wet sensitivity, and degree-day developmental profiles to give nearly identical climate suitability maps by CLIMEX modeling [4]: only this ref. 4, is not enough to declare above sentence. Please use appropriate reference.

Additional reference added and sentence was simplified (lines 58-59).

g [4]. These projections indicate that in the United States permanent SBL and FAW populations: check the space

Space deleted (line 60)

Is fall armyworm can overwinter in the TX? Which stage of FAW overwinter in the Tx and Canada?

FAW can overwinter in southern TX and includes all stages (full generation). Text and reference are added to note this (lines 60-62).

FAW density from bat stomach contents that had been feeding at high altitudes [7-10]. Modeling verified by genetic studies demonstrated that the Texas: check the space

 Space deleted (line 77)

Mountains potentially acting as a physical barrier that impedes more extensive interactions. As a result, the Texas and Florida wintering populations have limited interactions,: check the space

 Space deleted (line 83)

I found many writing errors in the manuscript, particularly with the space. Please check carefully and correct accordingly.

Editing was done as suggested.

Cytochrome oxidase subunit I, Triosephosphate isomerase: gene name should be italicized

Done (lines 120-123)

Why author used old samples? If possible, please compare with latest samples

The old samples (pre-2015) were for FAW and represents previous published data as indicated in the text (lines 308-309). Two references were added to make that clearer. The FAW data are used for comparison purposes. The SBL collections are all post-2015.

Please compare the mean differences for Figure 3B.

This was done as suggested in line 286-288 and in the Figure legend (line 301-303).

Round 2

Reviewer 1 Report

(refs) in Line 202 is an obvious problem.